# Healthcare-Associated SARS-CoV-2 Reinfection after 3 Months with a Phylogenetically Distinct Omicron Variant: A Case Report

**DOI:** 10.3390/v14091852

**Published:** 2022-08-23

**Authors:** Kim Callebaut, Anke Stoefs, Dimitri Stylemans, Oriane Soetens, Florence Crombé, Ellen Vancutsem, Hideo Imamura, Ingrid Wybo, Deborah De Geyter, Denis Piérard, Astrid Muyldermans, Thomas Demuyser

**Affiliations:** 1Department of Microbiology and Infection Control, Vrije Universiteit Brussel (VUB), Universitair Ziekenhuis Brussel (UZ Brussel), Laarbeeklaan 101, 1090 Brussels, Belgium; 2Department of Pulmonology, Universitair Ziekenhuis Brussel (UZ Brussel), Laarbeeklaan 101, 1090 Brussels, Belgium; 3Brussels Interuniversity Genomics High Throughput Core (BRIGHTcore) Platform, Vrije Universiteit Brussel (VUB), Universitair Ziekenhuis Brussel (UZ Brussel), Laarbeeklaan 101, 1090 Brussels, Belgium

**Keywords:** COVID-19, reinfection, Omicron, healthcare-associated

## Abstract

This case report describes a 60-year-old female patient suffering from systemic sclerosis, for which she received immunomodulatory drugs. Her first SARS-CoV-2-positive nasopharyngeal sample was obtained in the emergency department, on 31 January 2022. Whole genome sequencing confirmed infection with Omicron BA.1.1. Her hospital stay was long and punctuated by many complications, including admission to the intensive care unit. At the beginning of April 2022, she started complaining of increased coughing, for which another SARS-CoV-2 RT-qPCR test was performed. The latter nasopharyngeal swab showed a strongly positive result. To support the theory of healthcare-associated reinfection, whole genome sequencing was performed and confirmed reinfection with Omicron BA.2. Since this patient was one of ten positive cases in this particular ward, a hospital outbreak investigation was performed. Whole genome sequencing data were available for five of these ten patients and showed a cluster of four patients with ≤2 small nucleotide polymorphisms difference.

## 1. Introduction

According to the World Health Organization (WHO), Severe Acute Respiratory Syndrome Coronavirus 2 (SARS-CoV-2), has infected over 500 million individuals and claimed over 6 million lives around the world throughout the COVID-19 pandemic so far [1] Studies have shown that 90–99% of patients that endured a SARS-CoV-2 infection produce a detectable immune response within 2–4 weeks post-infection [2,3]. The strength and duration of this immune response, with regard to the susceptibility for reinfection, is however not well understood [3]. Hall et al. reported an 84% lower risk of reinfection over a period of seven months following a primary infection [4]. A Danish study performed by Michlmayr et al. observed sustained protection against SARS-CoV-2 for over one year in the unvaccinated Danish population, but reduced protection was seen with the emergence of new variants, such as Omicron. Their results indicate that viral evolution plays an important role in the protection against reinfection, making the prediction of reinfection rates difficult [5]. Kai-Wang To et al. described the first SARS-CoV-2 reinfection, confirmed by whole genome sequencing [6]. Although there is an extensive number of COVID-19 cases in general, only a few case reports describe genetically confirmed reinfections.

Healthcare-associated infections (HAIs) are one of the three most commonly reported in-hospital adverse events [7]. Weiner et al. reported that approximately 87% of HAIs are caused by 15 microorganisms, with most of them being bacterial [8]. Although most HAI-causing organisms are bacteria, several investigators reported the impact of respiratory viruses on HAI [9,10]. Viral HAI outbreaks with influenza virus [11], human respiratory syncytial virus [12], and SARS-CoV-2 [13,14] are significant threats to hospitalized patients.

Here, we present a case report regarding healthcare-associated SARS-CoV-2 reinfection with a phylogenetically distinct Omicron variant after 9 weeks in a Belgian 720-bed tertiary care center Universitair Ziekenhuis UZ Brussel (UZ Brussel).

## 2. Results

### 2.1. Case Description

The patient is a 60-year-old female with a medical history of systemic sclerosis, for which she is receiving immunomodulatory therapy. She presented at the emergency department of UZ Brussel on the 31st of January 2022, four days after obtaining a positive COVID-19 self-test. She was not vaccinated for SARS-CoV-2. She complained of dyspnea and a productive cough that had worsened in the two days beforehand. A SARS-CoV-2 real-time semi-quantitative polymerase chain reaction (RT-qPCR) on a nasopharyngeal (NP) swab confirmed COVID-19 infection, with cycle threshold (Ct) values of 17.4 for the E-gene and 19.6 for the N2-gene (Xpert, Cepheid, Sunnyvale, CA, USA). This result was strongly positive, consistent with a high viral load (≥10^5^–<10^7^ RNA copies/mL), and suggestive for a recent infection and a probable contagious patient [15]. Her clinical examination and lab results were not alarming, after which she was discharged.

Five days later, she returned to the emergency department with complaints of asthenia, loss of appetite, and increasing dyspnea. The laboratory results displayed an inflammatory blood count, with a C-reactive protein of 143.5 mg/L and RT-qPCR on her second NP swab repeatedly showed a strong positive result (≥10^5^–<10^7^ RNA copies/mL) [15] with Ct values of 21.69 for the E-gene and 20.88 for the S-gene (Altona Diagnostics GmbH, Hamburg, Germany). A computed tomography scan showed increased density with a crazy paving pattern and consolidation of both lower lung lobes, indicative of COVID-19 pneumonia. The patient was admitted to the infectiology department. There, she suffered a quick general deterioration with an increasing need for oxygen and complications from her systemic sclerosis, including cutaneous manifestations, non-specific interstitial pneumonia, and offset esophageal dysmotility. She was transferred to the intensive care unit (ICU) on 8 February.

After a long and complicated ICU stay, she was transferred to the pulmonology department on 17^th^ of March, followed by a transfer to the revalidation ward. On the 3rd of April, she complained of a cough for which, according to hospital regulations, a new nasopharyngeal SARS-CoV-2 RT-qPCR was performed. This sample tested strongly positive (≥10^5^–<10^7^ RNA copies/mL) with the Altona RT-qPCR, with CT values of 18.76 for the E-gene and 17.57 for the S-gene [15]. Figure 1 presents the timeline of her hospitalization. Other patients in the revalidation ward tested positive around the same period as our patient. The combination of her new symptoms and high viral load, and a possible outbreak in the revalidation ward led to the suspicion of healthcare-associated reinfection. 

Her symptoms during her second infection were less severe, limited to dyspnea, and coughing. She was discharged after a total stay of four months in the hospital.

### 2.2. Healthcare-Associated Outbreak Analysis

Different definitions are available for HAIs and according to the European center for disease prevention and control (ECDC) [16], our patient is classified as a definite HAI, since she was admitted for almost two months before developing new symptoms. While investigating the source of her infection, it became clear that nine other patients who shared a ward with our case patient had a positive SARS-CoV-2 RT-qPCR NF swab, as shown in Figure 2. Whole genome sequencing (WGS) data were available for five of our ten patients, all being an Omicron BA.2 variant. Minimal spanning tree (MST) analysis, Figure 3, showed clustering (≤2 small nucleotide polymorfisms (SNPs)) for four of these five patients, including our case patient, patient 7. Patient 6 did not cluster with these other patients with a difference of 12 SNPs compared to our small cluster.

All patients for whom no WGS data were available had a negative SARS-CoV-2 RT-qPCR until the end of March (patient 2–4) or at the beginning of April (patients 8–10), four days before testing positive, as illustrated in Figure 2. None of the patients shared a room, except for patients 7 and 8 who were placed in the same room for 6 days before patient 7 was diagnosed with COVID-19. Eight of our ten patients are classified as definite HAIs according to the ECDC case source definitions of COVID-19 [16].

### 2.3. SARS-CoV-2 Strain Analysis

The genome sequences of the SARS-CoV-2 viral strain at admission and during the patient’s latest episode of symptoms were compared using Nanopore sequencing (Oxford Nanopore Technologies, Oxford, UK). The average depths for specimen A and specimen B were 202 and 493, respectively, and missing bases for specimen A and specimen B were 192 and 227, respectively. Two genetically distinct variants were discovered: for the initial infection (specimen A—Figure 4A), an Omicron BA.1.1 variant was detected; and during the second episode (specimen B—Figure 4A), an Omicron BA.2. variant was detected, both using the pangolin and Nextclade lineage classification with high confidence. Both strains clearly differed, as shown in Figure 4A. They were located on different Omicron branches, as shown in Figure 4B.

These data strongly suggest reinfection with a phylogenetically distinct Omicron variant. Table 1 shows variant-specific base variations for Omicron BA.1-like strains and BA.2-like strains, and whether or not these are present or absent in our patient’s NP swabs.

### 2.4. SARS-CoV-2 Serology

Anti-nucleocapsid (anti-N) IgG and anti-spike protein (anti-S) IgG were determined on four samples collected during and after our patient was in hospital. The results are displayed in Table 2.

## 3. Materials and Methods

### 3.1. Healthcare-Associated Infections and Outbreak Analysis

We performed an outbreak analysis to trace the circulating HAI-associated viral strain. Ten COVID-19 patients were included in this analysis, as they were admitted to the same ward around the period of the case’s reinfection. ECDC defines COVID-19 infections in four different case sources. [16]

### 3.2. SARS-CoV-2 Reverse-Transcriptase Semi-Quantitative Polymerase Chain Reaction (RT-qPCR)

RNA extraction of the SARS-CoV-2 virus was carried out using the King Fisher Flex system (Thermo Fisher Scientific, Rocklin, CA, USA). The RT-qPCR was performed using the AltoStar^®^ SARS-CoV-2 RT-kit 1.5 (Altona Diagnostics GmbH, Hamburg, Germany) with primers targeting the E-gene and S-gene of the SARS-CoV-2 virus, on a CFX96 Deep Well Dx System real-time PCR cycler (Bio-Rad Laboratories, Hercules, CA, USA). A semi-quantitative estimation of the viral loads from Ct values was made according to Cuypers et al.; the interpretation and categorization of the semi-quantitative viral load were carried out accordingly [15]. Cuypers et al. categorized four different categories of SARS-CoV-2 positivity, ranging from very strongly positive, to strongly positive, followed by moderately positive and weakly positive, and these categories are directly linked to the SARS-CoV-2 viral load. Infectivity of these categories is interpreted as contagious, probably contagious, potentially contagious and, probably not contagious [15].

### 3.3. SARS-CoV-2 Whole Genome Sequencing (WGS)

RNA extraction of the SARS-CoV-2 virus was carried out using Maxwell extraction (Promega, Madison, WI, USA). WGS was performed using an adapted SARS-CoV-2 sequencing protocol [17] with the usage of Rapid Barcoding Kit 96 (SQK-RBK110.96). MinION R9.4.1 flow cells were used on the GridION x5 (Oxfors Nanopore Technologies, Oxford, UK) for the sequencing of both SARS-CoV-2 strains The viral reads were then analyzed with the ARTIC analysis pipeline 1.2.1 https://github.com/artic-network/fieldbioinformatics/ (accessed on 3 January 2022) [18]. Lineages were determined using the command-line pangolin version 4.1.1 with PUSHER-v1.11 [19]. Nextclade [20] Web 2.2.0 was used to create the viral base variation figure (Figure 4A) and Nextclade CLI 2.2.0 was also used for lineage analysis. UShER [21] was used to create the phylogenetic tree with existing relevant samples to observe the phylogenetic difference between two strains. The lineage-specific base variations used for Table 2 were obtained from pangolin version 4.1.1 data.

WGS data of our potential outbreak patients were processed with the SARS-CoV-2 plug-in of BioNumerics v.8.1 (Applied Maths, Biomérieux, Sint-Martens-Latem, Belgium). The subsequences of the Wuhan-Hu-1 (NC 04551219) reference genome were used as reference sequences for a BLAST search [22]. After extraction, these subsequences were screened for single-nucleotide polymorphisms (SNPs). Next, a similarity matrix was calculated based on the remaining SNP experiments and a minimal spanning tree (MST) was constructed. A genomic cluster is defined as genomes with ≤two SNPs difference.

### 3.4. SARS-CoV-2 Serology

Anti-N and anti-S IgG were determined with the Alinity i Anti-SARS-CoV-2 immunoassay (Abbott, IL, USA). Seroconversion for anti-S IgG was defined as values of ≥50 Arbitrary Unit (AU)/mL and an index greater than 1.4 for anti-N IgG.

All tests were performed in accordance with the manufacturer’s instructions.

## 4. Discussion

This case report presents a female patient who suffered from symptomatic reinfection with SARS-CoV-2. A few case reports were published on SARS-CoV-2 reinfection, such as a case from The Netherlands that describes an immunocompromised 89-year-old patient whose symptoms were worse during her second COVID-19 episode and eventually, she died of this reinfection [23]. Prado et al. described a 46-year-old male with more severe symptoms during his second episode including odynophagia, nasal congestion, fever, severe back pain, a productive cough, and dyspnea [24]. To et al. described a 33-year-old male patient who suffered asymptomatic reinfection with a distinct SARS-CoV-2 variant [6]. A case described by Van Elslande et al. talks about a 51-year-old female patient who suffered a mild second COVID-19 infection with symptoms such as headache, cough, and fatigue [25]. The symptoms of our patient were less severe during her second episode, indicating sufficient immunological memory, which is in contrast with the previously mentioned cases reported from The Netherlands [23] and Ecuador [24], but in line with other previous case reports from Belgium [25] and Hong Kong [6]. The protective immune response to infection with SARS-CoV-2 is not fully understood, but symptoms of reinfections in common cold and seasonal coronaviruses are usually less severe [26].

Studies have shown that 90–99% of patients that endured a SARS-CoV-2 infection produce a detectable immune response within 2–4 weeks post-infection [2,3]. It is unclear if our patient developed any antibodies during the period between both infections, since unfortunately, no serology testing was performed.

The occurrence of symptomatic reinfection only 3 months after the first infection with SARS-CoV-2 is not entirely unexpected. Multiple studies reported a short-lasting protective immunity in seasonal coronaviruses such as HCoV-NL63, HCoV-229E, HCoV-OC43, and HCoV-KHU1 with frequent reinfection within one year after the initial infection [27,28]. Previous case reports have reported genetically confirmed COVID-19 reinfection after 120, 140 and 68 days [24,25,29]. In our case report, reinfection occurred around day 70 after the primary infection.

Since the case patient suffered from a condition for which she was receiving immunomodulatory therapy, the question could arise as to whether the patient suffered from reinfection or prolonged shedding of SARS-CoV-2. The persistent presence of viral RNA or prolonged shedding can be a sequela of acute viral infections, with the immune clearance of the host considered as a major contributing factor [16]. This phenomenon was demonstrated for norovirus and influenza virus in individuals with compromised immune systems [30,31]. Various case reports have been published regarding the prolonged shedding of the virus in patients receiving immunomodulatory therapy [32,33]. Genetic analysis and knowledge of the evolution of SARS-CoV-2 can be of great help to determine whether a patient is suffering from prolonged shedding or whether the presence of virus particles is caused by a new, second infection. Luke et al. stated that multiple cases might share a linked transmission within a few days or weeks if there are fewer than approximately two SNP differences [34]. This is in accordance with the evolution speed reported by Tarhini et al. of 2–3 mutations per month [35]. The differences in base variations between specimens A and -B in our patient, as represented in Table 1, and the phylogenetic difference shown in Figure 4B are highly suggestive of reinfection, keeping in mind the evolution speed of the virus, rather than prolonged shedding in an immunocompromised patient.

SARS-CoV-2 serology testing showed that our patient did not develop anti-N and anti-S immunoglobulins 14, 37 and 157 days post-hospital admission. As mentioned previously, this case describes a patient receiving immunomodulatory drugs, more specifically mycophenolate mofetil (MMF), rituximab, and methylprednisolone. Her MMF therapy was temporarily put on hold at admission and her methylprednisolone was switched to dexamethasone. MMF can significantly reduce the rate of seroconversion in COVID-naïve patients, and the neutralizing capacity of anti-SARS-CoV-2 antibodies, according to a study performed by De Santis et al. [36]. Another study performed by Boekel et al. describes an absence in antibody response in 62.1% of rituximab-treated patients [37]. The absence of antibodies could be an indication of a diminished immune response for both of her COVID-19 episodes. A diminished immune response is, however, in contrast with the lower severity of the second infection. It is important to keep in mind that the immune system and immunity are not only related to antibodies created by the so-called humoral immune response, but that there is also T-cell-mediated immunity [38]. It is known from vaccine studies that rituximab has the potential to suppress the humoral immune system, leading to the absence of antibody formation [39,40,41]. However, in these patients the cellular immune response, mediated by T-cells, can be active, potentially giving the patient some protection for following infections [39,40].

A threshold of ≤two SNPs was used in our hospital outbreak investigation on the available WGS data to detect clusters and support the theory of an outbreak in the revalidation ward, eventually leading to the reinfection of our case patient. The presence of a cluster (Figure 3) leads to the probable assumption of an outbreak with an Omicron BA.2 variant. Patient 6, for whom WGS data showed an SNP difference of 12 with our cluster, was suffering from COVID-19 for more than ten days before the nine other patients started showing symptoms and had a positive SARS-CoV-2 RT-qPCR NF swab. Besides the SNP difference, patient 6 was still in isolation for COVID-19 at the time of the outbreak. To our knowledge, there were no confirmed infections in healthcare workers in this ward.

We should mention that limited contact tracing was performed in this outbreak investigation, mainly keeping the focus on our own patient; this is a possible limitation in the study.

## 5. Conclusions

Reinfection with a phylogenetically distinct SARS-CoV-2 Omicron variant can occur as soon as after 3 months. This case illustrates the risk for healthcare-associated reinfections and small outbreaks among critical patients, in this case in the revalidation ward, during their stay in the hospital. The clinical course of this patient’s second infection was less severe than during the first episode but otherwise has been described in the literature.

## Figures and Tables

**Figure 1 viruses-14-01852-f001:**
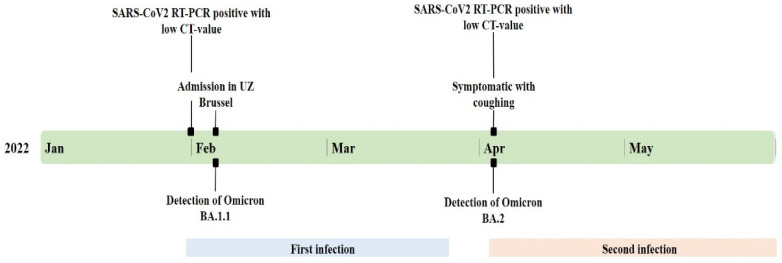
Timeline of patient’s hospitalization, including symptoms and genetic results; Ct: SARS-CoV-2 RT-qPCR cycle threshold.

**Figure 2 viruses-14-01852-f002:**
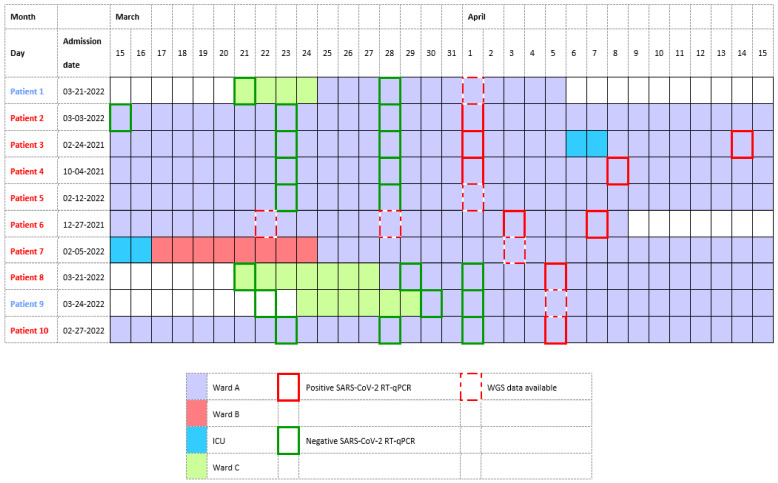
Epidemiological timeline of revalidation ward (ward A) cluster. All patients were admitted for more than seven days before their positive SARS-CoV-2 RT-qPCR NP swab (red box), and all had recent negative tests (green box), except for one (patient 6). WGS data were available for five of these ten patients, including this case patient (red dotted box). The patients mentioned in red are classified as definite HAI and those in blue as probable HAI according to the ECDC case source definitions of COVID-19.

**Figure 3 viruses-14-01852-f003:**
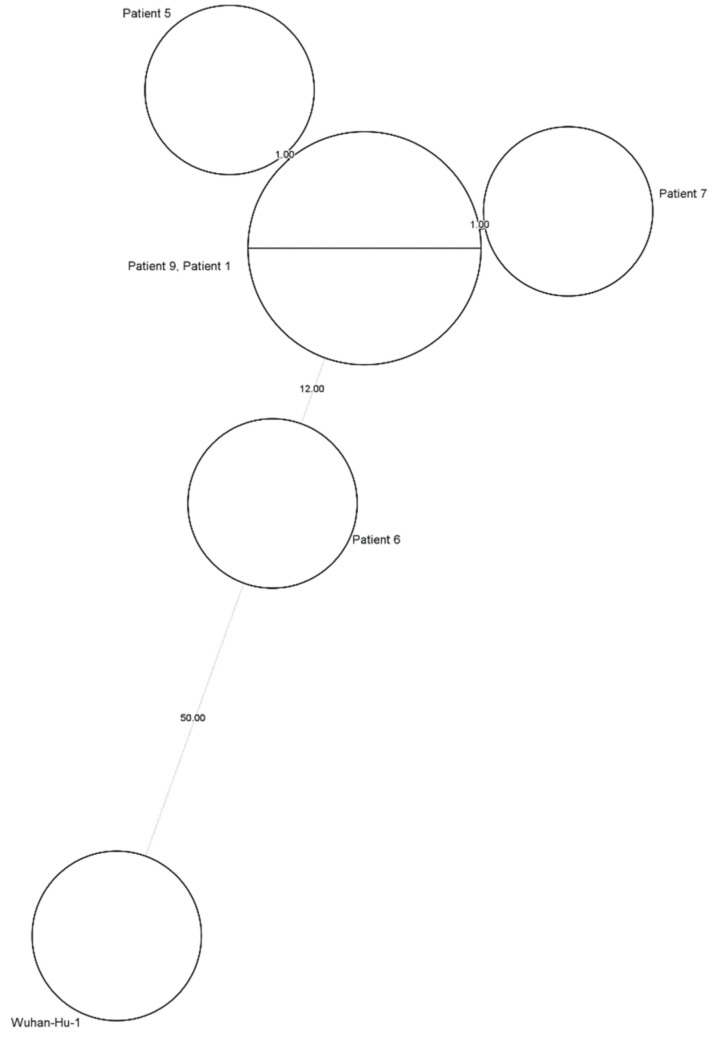
MST analysis of a possible outbreak cluster in 5 patients, with the Wuhan-Hu-1 as reference genome (NC 04551219).

**Figure 4 viruses-14-01852-f004:**
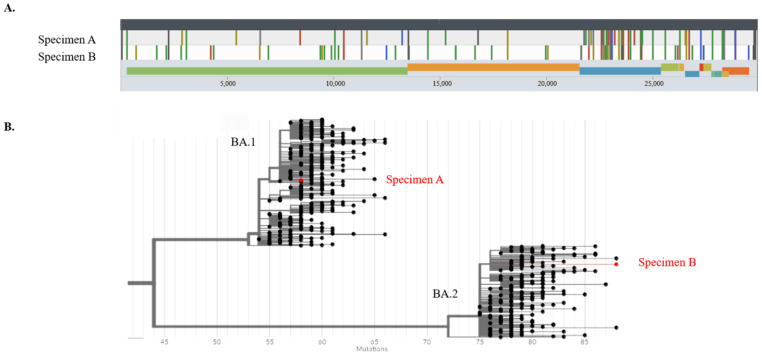
(**A**). Base variation plots of SARS-CoV-2 strains of the first (specimen A) and second (specimen B) coronavirus disease infections against the reference genome (**B**). A phylogenetic tree of Omicron BA.1 and BA.2 strains with specimens A and B generated by UShER. The phylogenetic tree is rooted at the Wuhan-Hu-1 reference and the numbers at the bottom indicate the base difference from the reference genome.

**Table 1 viruses-14-01852-t001:** Characteristic base variations, taken from pangolin version 4.1.1 data, for BA.1-like and BA.2-like strains, and their presence or absence in specimens A and B.

BA.1 Like Characteristic Base Variations			BA.2 Like Characteristic Base Variations		
	A	B		A	B
nuc:T5386G	+	−	ORF1ab:S135R	−	+
del: 6513:3	+	−	ORF1ab:T842I	−	+
ORF1ab:A2710T	+	−	ORF1ab:G1307S	−	+
ORF1ab:L3674F	−	−	nuc:C4321T	−	+
ORF1ab:SGF3675-	−	−	ORF1ab:L3027F	−	+
ORF1ab:I3758V	+	−	nuc:A9424G	−	+
nuc:T13195C	+	−	ORF1ab:T3090I	−	+
nuc:C15240T	+	−	ORF1ab:L3201F	−	+
S:A67V	+	−	ORF1ab:SGF3675-	−	+
del:21765:6	+	−	nuc:C10198T	−	+
S:T95I	+	−	nuc:G10447A	−	+
del:21987:9	+	+	nuc:C12880T	−	+
del:22194:3	+	−	nuc:C15714T	−	+
nuc:22205 + GAGCCAGAA	+	−	ORF1ab:R5716C	−	−
S:S371L	+	−	ORF1ab:T6564I	−	−
S:G446S	+	−	nuc:A20055G	−	+
S:Q493R	+	+	S:T19I	−	+
S:G496S	+	−	del:21633:9	−	+
S:T547K	+	−	nuc:T22200G	−	+
S:N856K	+	−	S:S371F	−	+
S:L981F	+	−	S:T376A	−	+
nuc:C25000T	+	+	S:D405N	−	+
nuc:C25584T	+	+	S:R408S	−	+
M:D3G	+	−	S:Q493R	+	+
N:RG203KR	+	+	nuc:C25000T	+	+
			nuc:C25584T	+	+
			nuc:C26060T	−	+
			nuc:C26858T	−	+
			ORF6:D61L	−	+
			N:RG203KR	+	+
			N:S413R	−	+

**Table 2 viruses-14-01852-t002:** Overview of SARS-CoV-2 antibody testing results.

Day *	Anti-Nucleocapsid IgG	Anti-Spike Protein IgG
0	Negative	Negative
14	Negative	Negative
37	Negative	Negative
157	Negative	Negative

* day since admission.

## Data Availability

Data of patients included in this manuscript are considered sensitive and will not be shared. The study methods are described in detail in Section 3. Virus genomic data are shared on GISAID (patient 1: EPI_ISL_11901621; patient 5: EPI_ISL_11901622; patient 6: EPI_ISL_11635206; patient 7: EPI_ISL_11901628 and EPI_ISL_10334126; patient 9: EPI_ISL_11901641).

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
