# Peer review of "Healthcare-Associated SARS-CoV-2 Reinfection after 3 Months with a Phylogenetically Distinct Omicron Variant: A Case Report"

_viruses, 2022, doi:10.3390/v14091852_

Round 1

Reviewer 1 Report

Authors report a case of Covid-19 healthcare associated reinfection. Less than 3 months occurred between the two infections. Nasal swabs were acquired and virus genome sequences were generated, which indicated that the two infections were caused by different omicron sub-variants, BA.1.1 and BA.2. Then an investigation is done, to characterize an outbreak that coincided with the second SARS-CoV-2 infection. It is an interesting and straightforward study, and it is well presented. I have a few minor recommendations:

-       Introduction: Authors could also mention the difficulty of estimating the rate of re-infection due to the constant evolution of the virus; susceptibility for reinfection will vary based on the circulating variants as well as previous exposure variant.  

-       It is unfortunate that not serological test was performed. As authors discuss, the patient was under immunomodulatory treatment for systemic sclerosis. This treatment might have interfered with the generation of antibodies during the first infection. Thus, it is possible that this patient was more susceptible to reinfection that other populations. Immunosuppression and serological responses are mentioned in the discussion, but not connected. Information about the specific immunomodulatory treatment is not specified. Is this information available? If it is, it would be interesting to include it in the study, and discuss if it is known if that therapy interferes with generation of adaptive immune responses.

-       Table 3 is “copied from” a different publication, which reference is provided. Since this is published, copying the table here does not seem appropriate. It would be better just to cite it and provide a brief description of each category. Detail information also does not seem necessary for interpretation of this study.

-       Line 85: “Her clinical examination and lab results were reassuring”; is reassuring here the right word? Please revise.

Author Response

Thank you for taking the time to assess our manuscript and for your feedback. The suggestion has been useful and we appreciate the opportunity to revise our manuscript. We tried to address your suggestion to the best of our possibility and you can find the revised manuscript as an attachment.

POINT 1: Introduction: Authors could also mention the difficulty of estimating the rate of re-infection due to the constant evolution of the virus; susceptibility for reinfection will vary based on the circulating variants as well as previous exposure variant.

Response: We agree with the reviewer on the importance of the constant evolution of the virus. We found a study in the Danish population illustrating that the susceptibility for reinfection has changed with the rise of variants and added this in our introduction (Line 35-40).

POINT 2: It is unfortunate that not serological test was performed. As authors discuss, the patient was under immunomodulatory treatment for systemic sclerosis. This treatment might have interfered with the generation of antibodies during the first infection. Thus, it is possible that this patient was more susceptible to reinfection that other populations. Immunosuppression and serological responses are mentioned in the discussion, but not connected. Information about the specific immunomodulatory treatment is not specified. Is this information available? If it is, it would be interesting to include it in the study, and discuss if it is known if that therapy interferes with generation of adaptive immune responses.

Response: We agree that the inclusion of antibody testing would definitely improve the quality of this study. For this, we performed serology testing on four samples: from 05/02/2022 (admission), 19/02/2022, 15/03/2022 and 12/07/2022.

Anti-spike protein IgG and anti-nucleocapsid IgG were determined, using Alinity I SARS-CoV-IgG and Alinity I SARS-CoV-2 IgG II Quant. Both tests were negative on all four samples, indicating that our patient did not have circulating antibodies. We did not perform a neutralization assay because the Alinity i SARS-CoV-IgG II Quant test, designed to detect immunoglobulins class G, including neutralizing antibodies to the RBD of the S1 unit of the spike protein, was negative. The results were included in the study:

Material and methods: Line 188-192

Results: Line 113-118, Table 2

Discussion: Line 238-255

POINT 3: Table 3 is “copied from” a different publication, which reference is provided. Since this is published, copying the table here does not seem appropriate. It would be better just to cite it and provide a brief description of each category. Detail information also does not seem necessary for interpretation of this study.

Response: We agree that copying the table is not the best solution. We adjusted this in the manuscript as suggested.

POINT 4: Line 85: “Her clinical examination and lab results were reassuring”; is reassuring here the right word? Please revise.

Response: we revised this as: “Her clinical examination and lab results were not alarming, after which she was discharged”. (Line 65-66)

We hope we addressed the request properly and that the revised manuscript will now better suit the special edition of Viruses: SARS-CoV-2 Research in Belgium. We would be happy to consider further revisions and thank you again for your time.

Reviewer 2 Report

The study by Callebaut et al. entitled “Healthcare-associated SARS-CoV-2 reinfection after 3 months with a phylogenetically distinct Omicron variants: a case report”, describes a 60-year-old female patient who has systemic sclerosis, for which she received immunomodulatory drugs. Her first SARS-CoV-2 positive nasopharyngeal sample was obtained in the emergency department, on January 31st, 2022. Whole genome sequencing confirmed infection with Omicron BA.1.1. Her hospital stay was extended and punctuated by many complications, including admission to the intensive care unit. At the beginning of April 2022, she started complaining of increasing cough, for which another SARS-CoV-2 RT-qPCR test was performed. The latter nasopharyngeal swab showed a strongly positive result. To support the theory of healthcare-associated reinfection, whole genome sequencing was performed and confirmed reinfection with Omicron BA.2. Since this patient was one of ten positive cases in this particular ward, a hospital outbreak investigation was completed. Whole genome sequencing data were available for five of these ten patients and showed a cluster of four patients with ≤2 small nucleotide polymorphisms difference. The study is important because investigates possible reinfection by two distinct SARS-CoV-2 Omicron variants. Thus, should be accepted after minor revisions.

Minor revisions

1) As this is a description of only one case, the authors must describe the exact date of the events and not mid-March or the beginning of April.

2) Page 7, line 195: remove “virus”, the acronym SARS-CoV-2 already contains the term virus.

3) There are a number of limitations that should be included in the article discussions:

3.1) This is the description of only one case;

3.2) The interval between infections of each variant and contact tracing was not performed properly;

3.3) There is no information on the titer of neutralizing antibodies after the first and second infection;

3.4) Authors should compare the cases of reinfection described in the literature, taking into account the sex and age of the patients, Ct value, time of evolution and viral dissemination, symptoms presented, and other factors.

Author Response

Thank you for taking the time to assess our manuscript and for your feedback. The suggestion has been useful and we appreciate the opportunity to revise our manuscript. We tried to address your suggestion to the best of our possibility.

POINT 1: As this is a description of only one case, the authors must describe the exact date of the events and not mid-March or the beginning of April.

We agree and revised this in the case description as suggested. (Line 58, Line 77-79)

POINT 2: Page 7, line 195: remove “virus”, the acronym SARS-CoV-2 already contains the term virus.

 We removed the word virus as suggested.

POINT 3: There are a number of limitations that should be included in the article discussions:

This is the description of only one case;

We agree that this can be considered as a limitation but since this paper is submitted as a case report we chose not to mention this limitation in the discussion.

The interval between infections of each variant and contact tracing was not performed properly;

We agree that only limited contact tracing was performed, during which we mainly focused on our own patient. This limitation was mentioned in the discussion (Line 264-267)

There is no information on the titer of neutralizing antibodies after the first and second infection;

Response: We agree that the inclusion of antibody testing would definitely improve the quality of this study and that this was a limitation in our case report. For this, we included samples from across patient admission 05/02/2022 (admission), 19/02/2022, 15/03/2022 and 12/07/2022.

Anti-spike protein IgG and anti-nucleocapsid IgG were determined, using Alinity I SARS-CoV-IgG and Alinity I SARS-CoV-2 IgG II Quant. Both tests were negative on all four samples, indicating that our patient did not form antibodies. We did not perform a neutralization assay because the Alinity i SARS-CoV-IgG II Quant test, designed to detect immunoglobulins class G, including neutralizing antibodies to the RBD of the S1 unit of the spike protein, was negative. We did however couple our results to our patient’s immunomodulatory therapy in the discussion.

The results were included in the study:

Material and methods: Line 188-192

Results: Line 113-118, Table 2

Discussion: Line 238-255

POINT 4: Authors should compare the cases of reinfection described in the literature, taking into account the sex and age of the patients, Ct value, time of evolution and viral dissemination, symptoms presented, and other factors.

We added more details, such as age, sex and symptoms in the discussion where our case is compared with other cases described in literature. We did chose not to compare Ct values, as different methodologies are used in different laboratories; we believe that exact Ct values are less informative than a semi-quantitative estimation of the log copies/mL. This semi-quantitative estimation is, however, frequently unavailable. (Line 195-204)

We hope we addressed the request properly and that the revised manuscript will now better suit the special edition of Viruses: SARS-CoV-2 Research in Belgium. We would be happy to consider further revisions and thank you again for your time.

Reviewer 3 Report

The authors reported a case of SARS-CoV-2 reinfection. The design of this study is ok, the results is reliable. However, there are several similar SARS-CoV-2 reinfection case studies published. This report is quite similar with previous published studies. I just doubt the novelty of repeat this kind of works. If the authors can move forward to detect the immune response, such as SARS-CoV-2 antibody (including neutralization antibody) that will improve the quality of this study. Thanks!

Author Response

Thank you for taking the time to assess our manuscript and for your feedback. The suggestion has been useful and we appreciate the opportunity to revise our manuscript. We tried to address your suggestion to the best of our possibility.

POINT 1: The authors reported a case of SARS-CoV-2 reinfection. The design of this study is ok, the results is reliable. However, there are several similar SARS-CoV-2 reinfection case studies published. This report is quite similar with previous published studies. I just doubt the novelty of repeat this kind of works. If the authors can move forward to detect the immune response, such as SARS-CoV-2 antibody (including neutralization antibody) that will improve the quality of this study. Thanks!

Response: We agree that the inclusion of antibody testing would definitely improve the quality of this study. For this, we included samples from across patient admission 05/02/2022 (admission), 19/02/2022, 15/03/2022 and 12/07/2022.

Anti-spike protein IgG and anti-nucleocapsid IgG were determined, using Alinity I SARS-CoV-IgG and Alinity I SARS-CoV-2 IgG II Quant. Both tests were negative on all four samples, indicating that our patient did not form antibodies. We did not perform a neutralization assay because the Alinity i SARS-CoV-IgG II Quant test, designed to detect immunoglobulins class G, including neutralizing antibodies to the RBD of the S1 unit of the spike protein, was negative. We did however couple our results to our patient’s immunomodulatory therapy in the discussion.

The results were included in the study:

Material and methods: Line 188-192

Results: Line 113-118, Table 2

Discussion: Line 238-255

We hope we addressed the request properly and that the revised manuscript will now better suit the special edition of Viruses: SARS-CoV-2 Research in Belgium. We would be happy to consider further revisions and thank you again for your time.

Round 2

Reviewer 3 Report

Thank the authors for detecting the immune response. The authors answered all the comments properly. The results are reliable and well discussed. I suggest publishing this works. Thanks!